# Development of Polysulfone Membrane via Vapor-Induced Phase Separation for Oil/Water Emulsion Filtration

**DOI:** 10.3390/polym12112519

**Published:** 2020-10-29

**Authors:** Nafiu Umar Barambu, Muhammad Roil Bilad, Mohamad Azmi Bustam, Nurul Huda, Juhana Jaafar, Thanitporn Narkkun, Kajornsak Faungnawakij

**Affiliations:** 1Department of Chemical Engineering, Universiti Teknologi PETRONAS, Bandar Seri Iskandar 32610, Perak, Malaysia; barambunafiu@gmail.com (N.U.B.); azmibustam@utp.edu.my (M.A.B.); 2HICoE-Centre for Biofuel and Biochemical Research, Institute of Self-Sustainable Building, Universiti Teknologi PETRONAS, Seri Iskandar 32610, Perak, Malaysia; 3Faculty of Food Science and Nutrition, Universiti Malaysia Sabah, Jalan UMS, Kota Kinabalu 88400, Sabah, Malaysia; 4Advanced Membrane Technology Research Center (AMTEC), School of Chemical and Energy Engineering (SCEE), Universiti Teknologi Malaysia (UTM), Johor Bahru 81310, Johor, Malaysia; juhana@petroleum.utm.my; 5National Nanotechnology Center (NANOTEC), National Science and Technology Development Agency (NSTDA), 111 Thailand Science Park, Pathum Thani 12120, Thailand; thanitporn.nar@ncr.nstda.or.th (T.N.); kajornsak@nanotec.or.th (K.F.)

**Keywords:** oil/water emulsion, environmental pollution, membrane development, polyethylene glycol, vapor-induced phase separation

## Abstract

The discharge of improperly treated oil/water emulsion by industries imposes detrimental effects on human health and the environment. The membrane process is a promising technology for oil/water emulsion treatment. However, it faces the challenge of being maintaining due to membrane fouling. It occurs as a result of the strong interaction between the hydrophobic oil droplets and the hydrophobic membrane surface. This issue has attracted research interest in developing the membrane material that possesses high hydraulic and fouling resistance performances. This research explores the vapor-induced phase separation (VIPS) method for the fabrication of a hydrophilic polysulfone (PSF) membrane with the presence of polyethylene glycol (PEG) as the additive for the treatment of oil/water emulsion. Results show that the slow nonsolvent intake in VIPS greatly influences the resulting membrane structure that allows the higher retention of the additive within the membrane matrix. By extending the exposure time of the cast film under humid air, both surface chemistry and morphology of the resulting membrane can be enhanced. By extending the exposure time from 0 to 60 s, the water contact angle decreases from 70.28 ± 0.61° to 57.72 ± 0.61°, and the clean water permeability increases from 328.70 ± 8.27 to 501.89 ± 8.92 (L·m^−2^·h^−1^·bar^−1^). Moreover, the oil rejection also improves from 85.06 ± 1.6 to 98.48 ± 1.2%. The membrane structure was transformed from a porous top layer with a finger-like macrovoid sub-structure to a relatively thick top layer with a sponge-like macrovoid-free sub-structure. Overall results demonstrate the potential of the VIPS process to enhance both surface chemistry and morphology of the PSF membrane.

## 1. Introduction

Oil and gas industries remain the major supplier of energy for continuous economic and industrial development [1]. However, the discharge of improperly treated oil/water emulsion generated during oil and gas production and processing remains the major concern [2]. This issue leads to the global implementation of wastewater treatment, reuse, and discharge limits imposed by regulations [3,4]. Conventional techniques for the treatment of oil/water emulsion such as dissolved air floatation, adsorption, coagulation, centrifugation, and many others are inefficient for the separation of small-size oil droplets (<10 µm) from water in the form of stable oil/water emulsion [5]. More importantly, they generate secondary pollutants, apart from the high cost and complexity of their operations [6,7]. Those challenges render them incapable of meeting the imposed discharge limits, as well as the economic sustainability [8,9].

Membrane-based technology has emerged as a promising option for the treatment of challenging oil/water emulsion wastewater due to its high separation efficiency, and economic and environmental advantages [10,11]. The technology is in a state of rapid development mainly to overcome the challenges imposed by membrane fouling [12,13]. Membrane fouling causes an increase in hydraulic resistance (flux decline) due to pore blockage and/or cake layer deposition [14,15]. To restore the membrane performances, frequent chemical cleaning is often applied. However, apart from the health and environmental risk of the chemicals, they also shorten the membrane lifetime (membrane degradation) [16], affecting both capital and operational expenditures [17]. Nonetheless, it can be properly addressed by controlling the oil droplets/membrane surface interaction via either surface modification [4] and/or membrane surface patterning [12,18]. The former involves the improvement in the membrane surface wetting property (hydrophilicity), to facilitate repellence of the hydrophobic oil droplets and hinder the formation of the cake layer and/or pore blockage [3]. The latter involves creation of macro/micro patterns on the membrane surface that promote turbulence flow of the oil droplets induced by fluid eddies to drive the oil droplets away from the membrane surface [12].

Membrane surface chemistry modifications have gained significant research interest due to their effectiveness, facile approach, and the availability of many optional additives [19,20]. Additives such as polyethylene glycol (PEG) and PEG-containing materials are attractive due to their abundance in carboxyl and alcohols groups, high availability, and their low cost [21]. They can be incorporated into the membrane material via surface grafting, surface coating, or in-situ surface segregation techniques [21,22].

Surface grafting suffered from difficulties in controlling the polymerization parameters for proper grafting density, as well as high cost and environmental sustainability [23]. On the other hand, surface coating suffers from poor adhesion of the hydrophilic-coated layer, which make it vulnerable from peeling off over time, apart from the cost and environmental sustainability issue [19]. However, in-situ surface segregation has recently been adopted by many researchers due to its economic and environmental sustainability [23]. It involves the blending of the additive with the membrane base polymer during dope solution preparation [21,24].

Zhu et al. [21] developed a hydrophilic polysulfone (PSF) membrane using poly(ethylene glycol) methacrylate (PEGMA) as an additive via the surface segregation technique. They reported a water flux of up to 110 (L·m^−2^·h^−1^) for treatment of 500 ppm of polyethylene glycol solution. However, the technique is largely restricted by the excessive additive leached out during the phase separation [25]. As the additive is hydrophilic, it rapidly migrated to the nonsolvent (water) bath, leaving behind a small amount entrapped within the membrane matrix after solidification [21]. Therefore, an effective strategy is required to hinder the excessive additive leached out, thus improving its density within the membrane matrix and/or surface for improved membrane performances [23]. However, controlling membrane formation offers a facile approach to overcome this challenge. It involves manipulation of both kinetic and thermodynamic properties of the cast film [26,27].

Nonsolvent-induced phase separation (NIPS) and vapor-induced phase separation (VIPS), or sometimes called vapor-nonsolvent-induced phase separation (V-NIPS), have been proved to offer an additional degree of freedom in controlling the resulting membrane properties [28]. The former involves immediate immersion of the cast film into the nonsolvent bath that typically leads to the formation of a porous top layer supported by a finger-like, highly porous macrovoid asymmetric sub-layer structure [29]. The latter involves slow absorption of the nonsolvent (water vapor) by the cast film when exposed to humid air, which leads to the formation of a top skin layer and sponge-like symmetrical sub-structure [30]. The VIPS method offers more control over the membrane formation. The slow nonsolvent penetration allows the formation of an immobile top layer that can restrict rapid PEG chains leached out during the nonsolvent immersion step [21]. It then facilitates entrapment of the additive within the membrane matrix and can eventually increase the surface hydrophilicity [23].

Zhu et al. [21] developed a hydrophilic PSF membrane via the V-NIPS technique on the PSF/PEGMA/DMAc system. They found that the PEGMA surface density shifted regularly by adjusting the cast film humid air exposure time. The water flux increased from 110 to 512 (L·m^−2^·h^−1^) by adjusting the humid air exposure time from 0 to 5 min. Such an improvement was attributed to the formation of a top immobile layer that trapped the PEGMA chains and restricted their migration into the nonsolvent bath during the subsequent immersion step. Dehban et al. [31] studied the effect of humid air exposure time on the performance of poly phenyl sulfone ultrafiltration membranes via the V-NIPS approach. They reported that pore size and permeate flux shifted regularly by adjusting the humid air exposure time. Water flux and mean pore radius increased from 17.12 to 20.79 (L·m^−2^·h^−1^) and from 0.96 to 1.44 nm, respectively, by adjusting the exposure time from 0 to 15 s.

To our best knowledge, the application of VIPS for the development of a hydrophilic PSF membrane from the PSF/PEG/DMAc system is limited and worth investigating. This study employed the VIPS approach to develop a hydrophilic PSF membrane using the PEG as an additive for oil/water emulsion filtration. Detailed surface structure and chemistry characterization of the membranes prepared by exposing the cast film to humid air at a range 0, 30, and 60 s were conducted to prove the hypothesis. The hydraulic and fouling resistance performances of the resulting membranes were evaluated by treating the oil/water emulsion feed in a crossflow filtration setup over five filtration cycles.

## 2. Materials and Methods

### 2.1. Membrane Preparation

The dope solution constitutes PSF (Mw = 78.0 × 10^3^ g/mol, Sigma Aldrich, St. Louis, MO, USA), dimethylacetamide (DMAc, 99.8% purity, Sigma Aldrich), lithium chloride (LiCl, Mw = 42.38 g/mol, ACROS Organics, Geel, Belgium), and PEG (Mw = 10,000 kg/mol, Sigma Aldrich) as polymer, solvent, pore former, and hydrophilic additive, respectively. The distilled water was used as the nonsolvent. The Novatexx 2413 (Freedenberg-filter, Weinheim, Germany) nonwoven was used as the baking support to avoid membrane shrinkage [32].

The homogeneously mixed dope solution was prepared by dissolving 18 wt% PSF, 1 wt% PEG, and 0.1 wt% LiCl (acting as pore former) in 80.9 wt% DMAc and stirred at 60 °C for 24 h. The solution was degassed and casted using a 0.22 mm wet casting thickness on the nonwoven support. Thereafter, the liquid film was exposed to humid air (70% relative humidity) at room temperature (of 22 °C) for 0, 30, and 60 s before immersion into the nonsolvent bath. The as-prepared membranes were denoted according to the exposure time range as PSF/PEG-0 (considered as the NIPS-based membrane), as well as PSF/PEG-30 and PSF/PEG-60 (both considered as VIPS-based membranes), respectively. They were stored in fresh nonsolvent when not in use. Figure 1 depicts the process flow diagram for membrane fabrications.

### 2.2. Feed Preparation

Stabilized oil/water emulsion was synthesized using crude oil (obtained from a crude oil well in Malaysia), distilled water, and sodium dodecyl sulfate (SDS, 98% purity, Sigma Aldrich) as the oil, water, and surfactant, respectively. The SDS-to-oil ratio of 1:99 (*w*/*w*) [33] was mixed in water to obtain 1000 ppm stabilized emulsion via mechanical agitation at a stirring rate of 3500 rpm for 24 h. The synthesized oil/water emulsion was stable for about 2 months without any sign of settling or oil floatation. A small volume of feed samples was later analyzed using a particle size and zeta potential analyzer (Malvern, Zetasizer Nano ZSP, Malvern, United Kingdom) to map the oil droplet size distribution.

### 2.3. Membrane Characterization

Field emission scanning electron microscopy (FESEM, ZEISS EVO ^®^ LS 15, Jena, Germany) was used to characterize the morphology of the prepared membranes. The samples were fractured in liquid nitrogen and coated with platinum prior to their surface and cross-sectional characterization with FESEM. A capillary flow porometer (Porolux 1000, Berlin, Germany) and goniometer (Ramé-Hart 260, Succasunna, NJ, USA) were used to determine the pore size distribution and contact angle of the membrane samples, respectively. The X-ray photoelectron spectrometer (XPS, K-Alpha^TM^, Thermo Scientific, Waltham, MA, USA) and Fourier-transform infrared spectrometer (FTIR, PerkinELmer, Inc., Waltham, MA, USA) were used to investigate the surface chemical compositions and surface functional groups, respectively. Finally, the oil concentration of the sample’s permeance was determined using a UV-VIS spectrometer (Shimadzu UV-2600, Kyoto, Japan) at a wavelength of 227 nm.

### 2.4. Filtration Setup

The filtration tests were run under a constant 0.2 bar trans-membrane pressure (Δ*P*) in a crossflow filtration setup, as depicted in Figure 2. A feed pump was installed in the system to continuously circulate the feed while a valve was installed to control the operating pressure measured through the pressure gauge, where Q_F_ is the feed flowmeter and was maintained at 418 mL·min^−1^, and Q_R_ and Q_P_ are the retentate and permeate flowmeter, respectively. Q_P_ data were used to evaluate the performance of the membrane over time, and their data have been provided in the results and discussion section. The membrane sample was cut and installed in the filtration cell with an effective area of 0.0037 m^2^ and compacted for 60 min prior to the clean water permeability (CWP) test.

The sequence of filtration tests was conducted as follows: After 60 min of membrane compaction, the flux reached a nearly constant value for three consecutive samples collected every 10 min and recorded as the CWP. After measurement, the permeate was recycled back to the feed to maintain the feed condition (i.e., oil concentration). Only a small volume of the permeate was kept for analysis. The feed was then switched to oil/water emulsion and the filtration was continued for 90 min while permeate samples were collected every 10 min, followed by water cleaning for 10 min prior to CWP re-evaluation. This procedure was regarded as one complete filtration cycle. The filtration was further continued for another four cycles to fully evaluate the performance of the membranes, as well as to evaluate the membrane fouling.

The CWP and the oil/water emulsion permeabilities (*L* (L·m^−2^·h^−1^·bar^−1^)) were calculated using Equation (1), while oil rejection performances (R, %) were calculated using Equation (2).
(1)$$L=\;\frac{\Delta V}{A\;\Delta t\;\Delta P\;}$$
(2)$$R=\left(1-\;\frac{C_{p}}{C_{f}}\right)\times \;100$$
where Δ*V* is the permeate volume collected (L), Δ*t* is the filtration time (h), Δ*P* is the trans-membrane pressure (bar), *A* is the membrane effective area (m^2^), *C_p_* is the permeate oil concentration (ppm), and *C_f_* is the feed oil concentration (ppm). The *C_p_* of the samples collected from the membranes were evaluated using a UV-VIS spectrometer (Shimadzu UV-2600, Kyoto, Japan) at a wavelength of 227 nm.

### 2.5. Membrane Fouling Resistance Test

The antifouling property of the membranes were analyzed by evaluating the fouling indices, namely permeance recovery (PR, %), reversible fouling (RF_,_ %), irreversible fouling (IrF, %), and residual permeability (RP, %) using Equations (3)–(6), respectively. The initial CWP is denoted as *CWP_wi_*_,_ while the CWP of the subsequent filtration cycles is denoted as *CWP_wi+n_*, in which *n* is the number of filtration cycles. Lastly, *F_oi_* denotes the oil/water emulsion permeability.
(3)$$PR=\;\frac{CWP_{wi+n}}{CWP_{wi}}\times 100\%$$
(4)$$RF=\frac{CWP_{wi+n}-F_{oi}}{CWP_{wi}}\;\times 100\%$$
(5)$$IrF=\frac{CWP_{wi}-CWP_{wi+n}}{CWP_{wi}}\;\times 100\%$$
(6)RP= PR− RF

## 3. Results and Discussion

### 3.1. Membrane Characteristics

#### 3.1.1. Morphology

The SEM images in Figure 3 show the influence of humid air exposure time on the resulting membranes morphologies. Clear changes in the membrane surface and cross-sectional morphologies are observed as a result of different fabrication conditions. The PSF/PEG-0 was prepared via the NIPS as the exposure time was set at 0 s by immediate immersion in the nonsolvent bath after casting. It has a typical structure normally exhibited by membranes fabricated via instantaneous demixing. The PSF/PEG-0 cross-sectional morphology shows an asymmetric structure with a thin skin on the top layer acting as the separation layer, supported by a highly porous finger-like sub-structure and a small portion of the sponge-like at the bottom [34,35]. By contrast, membranes formed by the delayed demixing mechanism show a sponge-like macrovoid-free sub-structure with a dense and thick skin layer.

The VIPS-based membranes have a somewhat symmetric cross-section morphology. Unlike the PSF/PEG-60 that has no macrovoid, the PSF/PEG-30 exhibits some macrovoids not fully suppressed at the sub-layer (Figure 3B). They are originated from the formation of a cluster of water-rich zones that rapidly enter the cast film through a pre-formed surface pore. The symmetric morphology of the VIPS-based membrane can be ascribed by the nature of membrane formation mechanisms. When exposed to humid air, the uptake of water could cause phase separation in the upperpart of the cast film to form an immobile structure of the membrane [24,34]. During immersion into the nonsolvent bath to complete the phase separation, it only slightly altered the overall structure of the film of PSF/PEG-30 to allow formation of a few macrovoids. The final structure of the PSF/PEG-60 membrane seemed to be formed during the exposure time to humid air. Even though the phase inversion was not completed, the cast film viscosity was so high that further exchange of the solvent and nonsolvent during the immersion did not change the overall structure of the formed membrane.

The surface SEM image of the PSF/PEG-0 clearly shows the surface pore within the range of microfiltration, further characterized in Section 3.1.2. Formation of the microfiltration range of the PSF membrane was facilitated by the presence of the PEG additive that promote instantaneous demixing [36,37]. Conversely, for the VIPS membranes, the pore sizes are much smaller and are not clearly visible from the surface SEM images. It is also worth noting that some concave shape can be seen on the PSF/PEG-30, most likely originating from the cast film that sank into the voids in the nonwoven support. A similar phenomenon most likely also occurred for the PSF/PEG-60, but the surface flattened given the prolonged time. Formation of a small surface pore size of the VIPS membrane occurred—as for the cross-sectional morphology—because of the pre-formation of the overall membrane structure during the exposure to humid air. The phenomenon of VIPS-based PSF membrane formation is slightly different than for the PVDF polymer. The pore size of VIPS-based PVDF membranes is typically larger than that of the NIPS-PVDF membranes [25,38].

The membrane thickness shows an increasing trend; as the exposure time raised from 0 to 30 and 60 s, the membrane thickness increased from 218.3 ± 1.3 to 234.3 ± 1.3 and 235.7 ± 1.7 μm, respectively. The slow uptake of water during the humid exposure of the cast film resulted in delayed demixing, which favors the pre-formation of an immobile top layer. Therefore, longer exposure time resulted in an increase in the thickness of the immobile top layer that eventually hindered the rapid exchange of solvent and nonsolvent when immersed in the nonsolvent bath [23,39]. Similar results were reported by Venault et al. [23] on the influence of humid air exposure time on the morphology and thickness of PSF membranes. The membrane was transformed from a finger-like to sponge-like sub-structure, as well as a gradual shift in the membrane thickness as a function of humid air exposure time.

#### 3.1.2. Mean Flow Pore Size and Pore Size Distribution

The mean pore size of the membranes gradually shifted to become smaller as the humid air exposure time was prolonged from 0 to 30 and 60 s, as illustrated in Figure 4. The NIPS-based PSF/PEG-0 exhibited a mean flow pore size of 0.126 μm, while the VIPS-based PSF/PEG-30 and PSF/PEG-60 had mean flow pore sizes of 0.057 and 0.032 μm, respectively. These results can be explained by considering both thermodynamic and kinetic effects of PEG and the humid air exposure time. Addition of PEG into the dope solution enhances the rate of liquid–liquid demixing due to the decrease in the polymer solution thermodynamic stability, while from the kinetic aspect, it slows down the liquid–liquid demixing rate due to the increase in the cast film viscosity. However, one of the effects supersede the other based on the phase separation condition [31]. For the NIPS-process, immediate immersion of the cast film favors instantaneous demixing with the presence of the hydrophilic additive that behaves like a nonsolvent [21]. Therefore, the NIPS-membrane (PSF/PEG-0) has a microfiltration range of pore size. In contrary to the VIPS-membrane, water uptake during exposure to humid air leads to an increase in cast film viscosity, to the extent of forming an immobile gel-like film. The high viscosity of the cast film resulted in delayed demixing that typically produced dense membranes or that of small pore size [25,35]. As the exposure time was extended, the viscosity of the cast film was increased. It was observed that the dense membrane was formed when the exposure time was extended beyond 60 s. Formation of a gel-like cast film prevented the further growth of nuclei during immersion in the nonsolvent bath resulting in a membrane with smaller mean flow pore size.

#### 3.1.3. Surface Hydrophilicity

The membranes surface hydrophilicity measured via the water contact angle decreases from the NIPS-membrane to the VIPS membrane and as the humid air exposure time was prolonged (Figure 5). By adjusting the exposure time from 0 to 30 s, the water contact angle decreased from 70.28 ± 0.61° to 67.09 ± 0.48°. By further adjustment to 60 s, the water contact angle decreased to 57.72 ± 0.61°. The decrease in water contact angle can be justified by the increase in PEG residue near the surface. This finding proves our hypothesis on the formation of an immobile structure of cast film during the exposure to humid air that entraps the PEG additive within the polymer matrix and reduces its leaking to the coagulation bath.

The finding of the decrease in water contact angle at longer exposure time can be attributed to an increase in PEG near the membrane surface, as also reported elsewhere [40], thanks to the immobile top layer that entrapped the PEG chains [35]. The entrapment of hydrophilic additive allows the formation of an hydration layer between water molecules and carboxylic groups, and alcohols groups of PEG chains that act as a physical barrier for oil droplets/membrane surface interaction [25]. Such a membrane is expected to pose better performance for the treatment of oil/water emulsion. In another study, Zhu et al. [21] also reported a gradual decrease in water contact angle for PEGylated PSF with the increase in humid air exposure time. By raising the exposure time from 0 to 1 and 3 min, the water contact angle decreases from 65° to 61° and 45°, respectively. In their approach, the PEG was cross-linked in situ.

#### 3.1.4. Fourier-Transform Infrared Spectra

Figure 6 illustrates the FTIR spectra of surface functional groups of the membranes, ascribing the presence of residual PEG in the VIPS-based PSF membrane qualitatively. For all the membranes, peaks attributed to the stretching vibrations of SO_2_ groups and aromatic rings that form the structural components of the PSF polymer were observed at 1150 and 1587 cm^−1^, respectively [20], while peaks ranging from 2800 to 3400 cm^−1^ and 1330 to 1420 cm^−1^, respectively, attributed to carboxylic and alcohols groups of PEG were also observed [41,42].

The peaks attributed to carboxylic and alcohols groups became more prominent by adjusting the humid air exposure time from 0 to 60 s. Moreover, the C–O stretching observed at 1050 and 1101 cm^−1^ also became more prominent for the VIPS-based membrane. This finding suggests that more PEG chains are trapped within the membrane matrix from the VIPS-based membrane. It ascribes the impact of the VIPS method in limiting the excessive leaching out of PEG during the immersion in the nonsolvent bath. Similar findings were also reported by Daramola et al. [20] and Zhu et al. [21] on the enhancement of PEG chains surface coverage with prolonged humid air exposure time. Thanks to the uptake of water from humid air, the preformation of the top immobile layer helps to trap the PEG chain within the membrane matrix and hinders its excessive leach out during the NIPS step. Eventually, it makes the membrane surface more hydrophilic [23].

#### 3.1.5. Surface Chemical Composition

The results of elemental composition from the Energy-dispersive X-ray spectroscopy mapping summarized in Table 1 show that the oxygen level increased by adjusting the humid air exposure time. It further confirms the entrapment of the PEG chain by application of the VIPS for PSF membrane fabrication. The oxygen element in Table 1 originates from the carboxylic and alcohols groups of the PEG. The increase in the oxygen content demonstrates the increasing amount of PEG entrapped in the membrane matrix, which was also promoted by the extent of the exposure time, as also reported by others [24,28]. By adjusting the exposure time from 0 to 30 s, the oxygen level increased from 24.64 to 25.86 wt%. It further increased to 26.26 wt% by adjusting the exposure time to 60 s. The finding further confirms the results in Figure 5 and Figure 6. A similar result was also reported by Nawi et al. [25], in which an increasing oxygen concentration from 2.22 to 3.25 wt% was found by adjusting the exposure time from 0 to 5 min, respectively, for VIPS-based PVDF membranes.

To further confirm the impact of exposure time on surface elemental compositions, two selected membrane samples (PSF/PEG-0 and PSF/PEF-60) were further characterized by XPS, as depicted in Figure 7. The result shows an increase in oxygen content for PSF/PEG-60. The surface concentration of O 1s increases from 16.99 to 17.96 mol% by adjusting the exposure time from 0 (NIPS-based membrane) to 60 s (VIPS-based membrane), respectively, while the O 1s to C 1s ratio increases from 0.208 to 0.229, as shown in Table 2. The findings suggest that more PEG chains cover the membrane surface by prolonging the cast film humid air exposure time, as also reported elsewhere [21].

#### 3.1.6. Clean Water Permeability

Figure 8 demonstrates the effect of humid air exposure time on the CWP. By adjusting the exposure time from 0 to 30 and 60 s, the CWP increases from 328.7 ± 8.27 (L·m^−2^·h^−1^·bar^−1^) to 364.78 ± 7.38 (L·m^−2^·h^−1^·bar^−1^) and 501.89 ± 8.92 (L·m^−2^·h^−1^·bar^−1^), respectively. This finding suggests an improvement in the hydration layer that enhances water molecules/membrane surface interactions, and further confirmed the results shown in Figure 5, Figure 6 and Figure 7. Dehban et al. [31] reported an increase in the CWP from 5.71 (L·m^−2^·h^−1^·bar^−1^) to 6.93 (L·m^−2^·h^−1^·bar^−1^) by adjusting the humid air exposure time from 0 to 15 s. It is worth noting that the low filtration resistance attributed to large pore size (Figure 4) is overshadowed by the hydration impact. Another plausible explanation is due to the higher number of pore population for the PSF/PEG-60 membrane that results in low intrinsic hydraulic resistance, as suggested by others [43].

### 3.2. Oil/Water Emulsion Filtration

Figure 9 shows that PSF/PEG-60 sustains up to 248.92 ± 10.54 (L·m^−2^·h^−1^·bar^−1^) permeability even after the long-term fouling test (90 min) with just 10 min of water cleaning, while PSF/PEG-0 and PSF/PEG-30 maintain only a permeability of 123.85 ± 1.82 and 169.46 ± 3.08 (L·m^−2^·h^−1^·bar^−1^) respectively. Even after the fifth filtration cycle, PSF/PEG-60 outperformed the rest by maintaining a permeability of up to 139.46 ± 0.81 (L·m^−2^·h^−1^·bar^−1^). On the other hand, PSF/PEG-0 and PSF/PEG-30 demonstrated only 45.16 ± 0.41 (L.m^−2^·h^−1^·bar^−1^) and 69.93 ± 0.28 (L·m^−2^·h^−1^·bar^−1^), respectively. The superior performance of the PSF/PEG-60 can be attributed to its properties detailed earlier. The hydrophilic nature of the surface facilitates a hydrogen bond with water to create a hydration layer that inhibits interaction of oil droplets with the membrane surface. The low pore size hinders the pore blocking by the oil droplets.

Figure 9 also shows that most of the permeability loss occurs during the first 10 min of the filtration, in which up to 50% of the permeability diminishes. This phenomenon can be ascribed by irreversible adsorption of the foulant that commonly occurs when treating a fouling-prone feed, as also discussed elsewhere [44,45]. Similarly, Nawi et al. [25] reported up to an 80% decline in permeability during the first 10 min of oil/water emulsion filtration, while Elhady et al. [46] reported a permeability loss of up to ~63% during oil/water emulsion filtration.

Figure 10 shows that the PSF/PEG-60 membrane demonstrates the highest oil rejection performance. By adjusting the humid air exposure time from 0 to 30 and 60 s, the oil rejection gradually improved from 85.06 ± 1.6% to 97.12 ± 2.2% and 98.48 ± 1.2%, respectively. This is attributed to the decrease in mean pore size with the increase in the humid air exposure time, and as such, the smaller oil droplets could be rejected [21]. Based on the particle size distribution of the oil droplets in the oil/water emulsion sample, the sizes of the droplets were distributed in a three-modal distribution at 0.25, 0.9, and 4.0 µm. Those droplet sizes are indeed way smaller than the mean flow pore size of all membrane samples (Figure 4). Another plausible explanation is the hydrophilic nature of the PSF/PEG-60 membrane in comparison to the rest, as also reported elsewhere [47]. It is anticipated that, due to severe fouling at higher oil concentration, the application of the membrane for oil/water emulsion filtration is only applicable up to a certain threshold value. An economic trade-off is then expected between the high concentration factor and the feasibility of the membrane, which will be addressed in a future study.

### 3.3. Membrane Fouling Resistance Analysis

Figure 11 shows that both VIPS-based membranes are better than the NIPS-based membrane. As shown in Figure 10, most of the permeability was lost during the first cycle, leaving a small range of the permeability for operation. However, after the third cycle, only a small increment in IrF is observed for all membranes. When comparing PFS/PEG-0 and PSF/PEG-60, the RF increases from 8.26 ± 0.95% to 19.45 ± 1.96%, respectively. On the other hand, the IrF decreases from 62.28 ± 1.5% to 50.43 ± 1.22%, respectively, whereas PR increases from 37.72 ± 1.5% to 49.57 ± 1.22%. The increase in RF and PR coupled with the decrease in IrF demonstrated an enhancement in fouling resistance, as suggested by others [48]. The fouling resistance enhancement can be attributed to the increased PEG surface density that continuously repelled the oil droplets and restricted their excessive interaction with the membrane surface (via adsorption) [49]. It is worth noting that substantial hydraulic performance was diminished during the first filtration cycle, suggesting the limitation still faced by the developed membrane. Almost 60% of irreversible fouling was suffered by all membranes. Under this condition, it substantially reduced the operability of the membrane system because of the intensive need of chemical cleaning to restore the irreversible fouling.

The higher performance demonstrated by PSF/PEG-60 can be attributed to the improved PEG surface density that enhances the surface hydrophilicity. PSF/PEG-0 has the least hydraulic and antifouling performances due to its lower surface hydrophilicity. Therefore, it has higher oil droplets/surface interaction that leads to the accumulation of higher oil droplets. The accumulation of oil droplets on the membrane surface leads to blockage of the membrane surface pores diminishing the hydraulic performance [25]. Similarly, Zhu et al. [21] reported the improvement in fouling resistance by adjusting the humid air exposure time from 0 to 5 min. They reported a decrease in the IrF from ~32.4% to 15.3% after filtration of a bovine serum albumin solution. On the other hand, the PR increases from ~67.6% to 84.5%.

This study demonstrates the application of the VIPS method to develop membranes for oil/water emulsion separation. The introduction of a simple step via exposure to humid air facilitates the formation of a more porous and more hydrophilic membrane suitable for oil/water emulsion separation. The developed membrane can be implemented for treatment of oil/water emulsion often found in produced water, palm oil mill effluent, etc. The enhanced hydraulic performance and better fouling resistance posed by PSF/PEG-60 would lead to a lower system footprint and simpler operation due to fewer actions required for membrane fouling management.

## 4. Conclusions

Overall results demonstrate that both the membrane structure, and hydraulic and fouling resistance performances are affected by adjusting the cast film humid air exposure time to undergo VIPS-based membrane synthesis. By adjusting the exposure time from 0 to 60 s, the membrane structure was transformed from a porous, thin top layer with a finger-like macrovoid sub-structure to a relatively thick top layer with a macrovoid-free sponge-like sub-structure. Moreover, the CWP and the PR increased from 328.7 ± 8.27 (L·m^−2^·h^−1^·bar^−1^) to 501.89 ± 8.92 (L·m^−2^·h^−1^·bar^−1^) and from 37.72 ± 1.5% to 49.57 ± 1.22%, respectively, while IrF decreased from 62.28 ± 1.5% to 50.43 ± 1.22%, respectively. The findings justified improvements in both hydraulic and fouling resistance performances, thanks to the preformation of the top immobile layer that trapped more PEG within the membrane matrix that hinders its excessive leach-out during the NIPS step, and makes the surface more hydrophilic and imposes the antifouling properties. As a result, it enhanced the water molecule/membrane surface interactions (formation of hydration layer) and restricted oil droplets/membrane surface interactions.

## Figures and Tables

**Figure 1 polymers-12-02519-f001:**
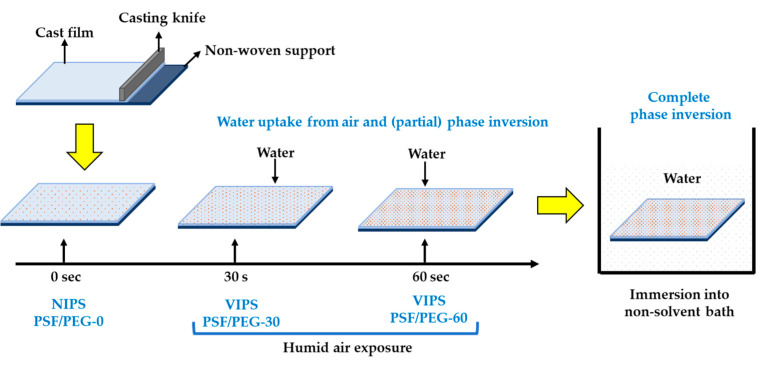
Illustration of membrane fabrication via vapor-nonsolvent-induced phase separation (V-NIPS) approach.

**Figure 2 polymers-12-02519-f002:**
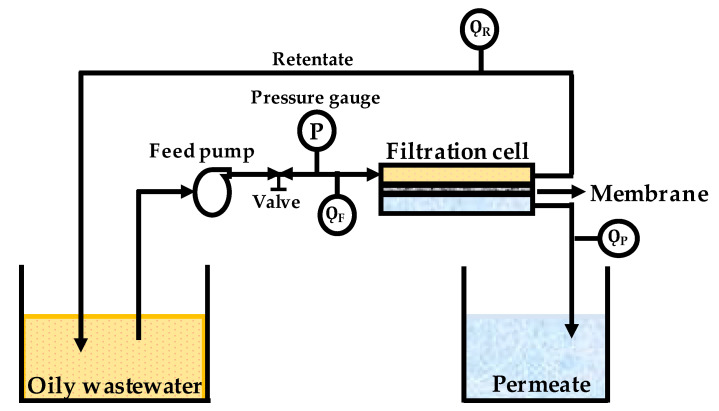
Crossflow filtration setup for oil/water emulsion filtration. After volume measurement, the permeate was turned to the feed contained to maintain the feed condition.

**Figure 3 polymers-12-02519-f003:**
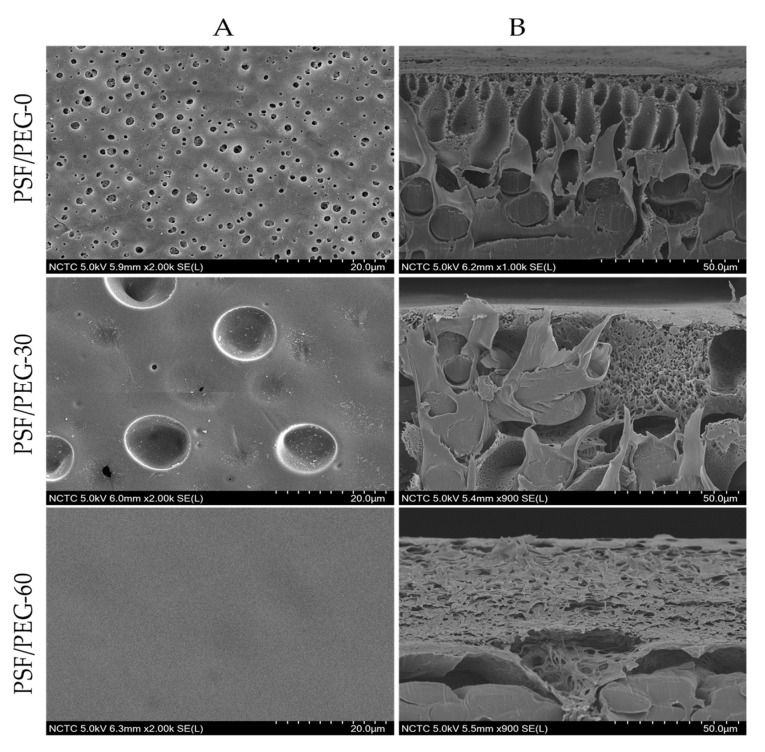
FESEM images of top surface (**A**) and cross-sectional (**B**) morphology of the membranes at 2000× and 900× magnification, respectively.

**Figure 4 polymers-12-02519-f004:**
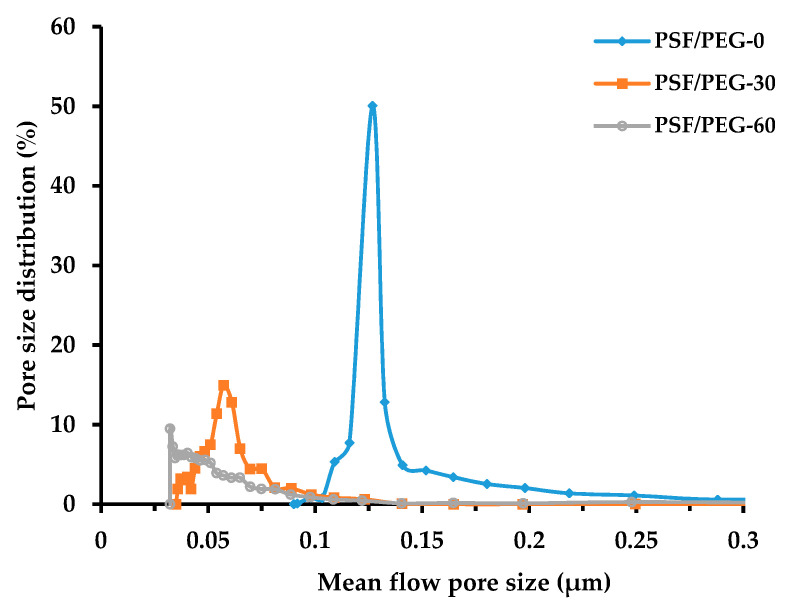
Pore size distribution of the resulting membranes.

**Figure 5 polymers-12-02519-f005:**
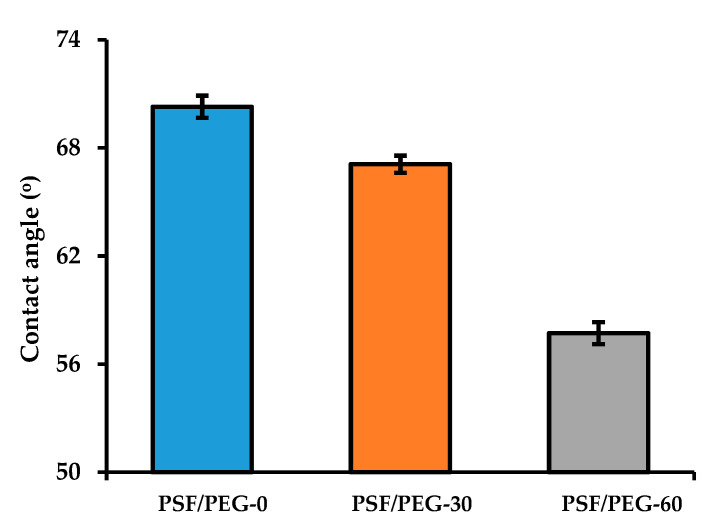
Water contact angle of the developed membranes.

**Figure 6 polymers-12-02519-f006:**
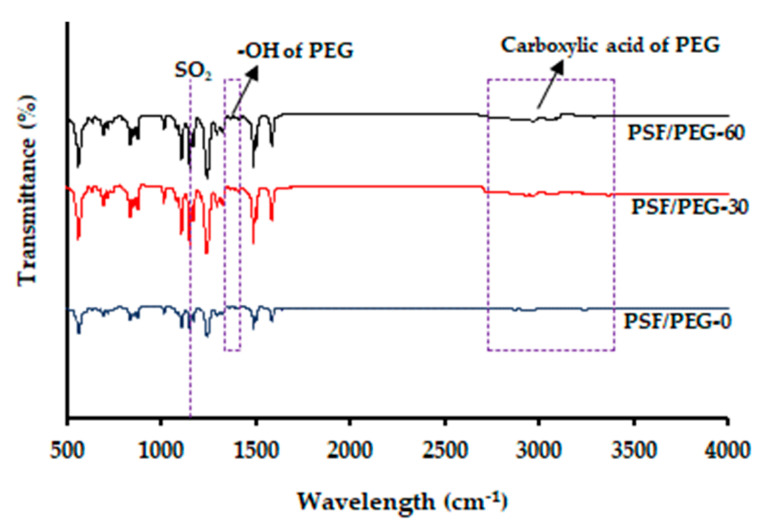
FTIR spectra of the membrane’s functional groups.

**Figure 7 polymers-12-02519-f007:**
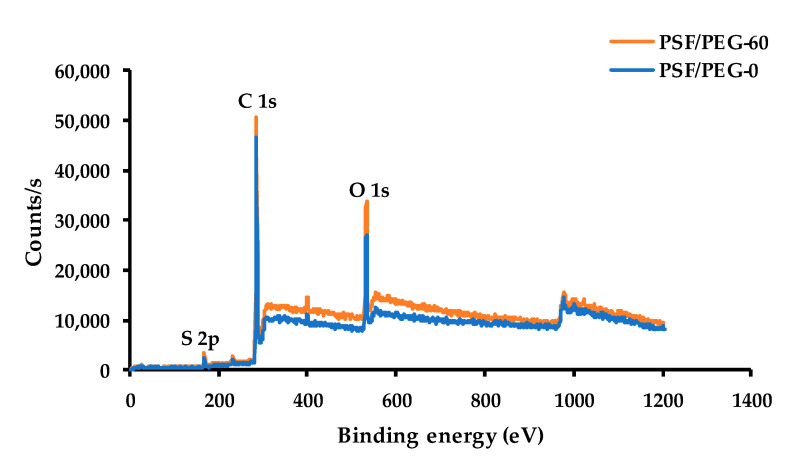
XPS wide-scan spectra of polysulfone (PSF)/PEG-0 and PSF/PEG-60 membranes.

**Figure 8 polymers-12-02519-f008:**
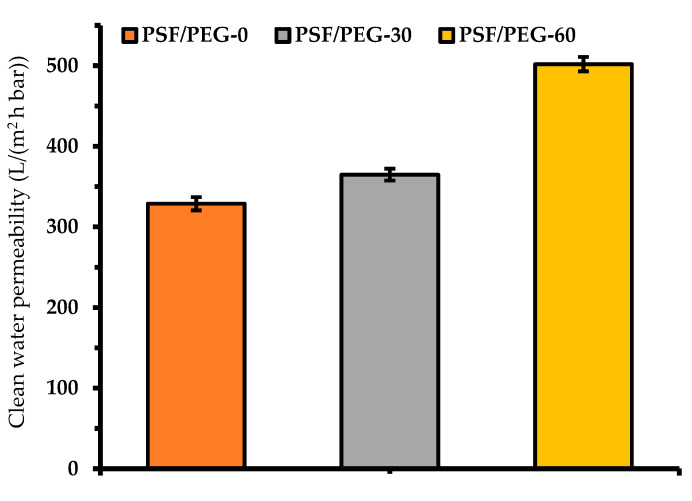
Clean water permeability of the resulting membranes.

**Figure 9 polymers-12-02519-f009:**
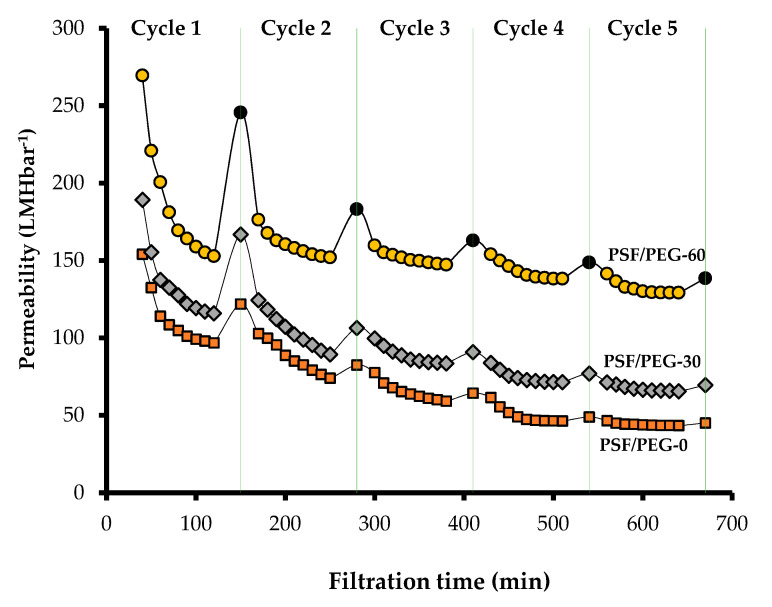
Hydraulic performance of the membrane for oil/water emulsion filtration.

**Figure 10 polymers-12-02519-f010:**
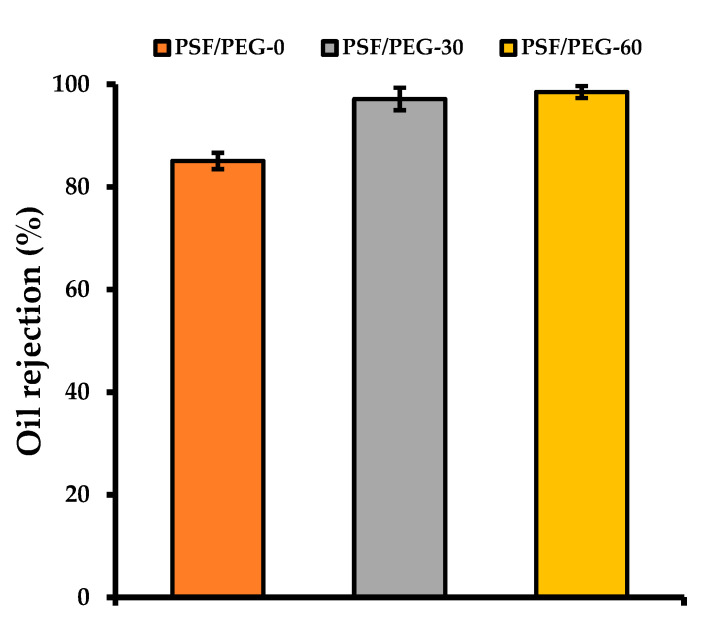
Effect of humid air exposure time on oil rejection.

**Figure 11 polymers-12-02519-f011:**
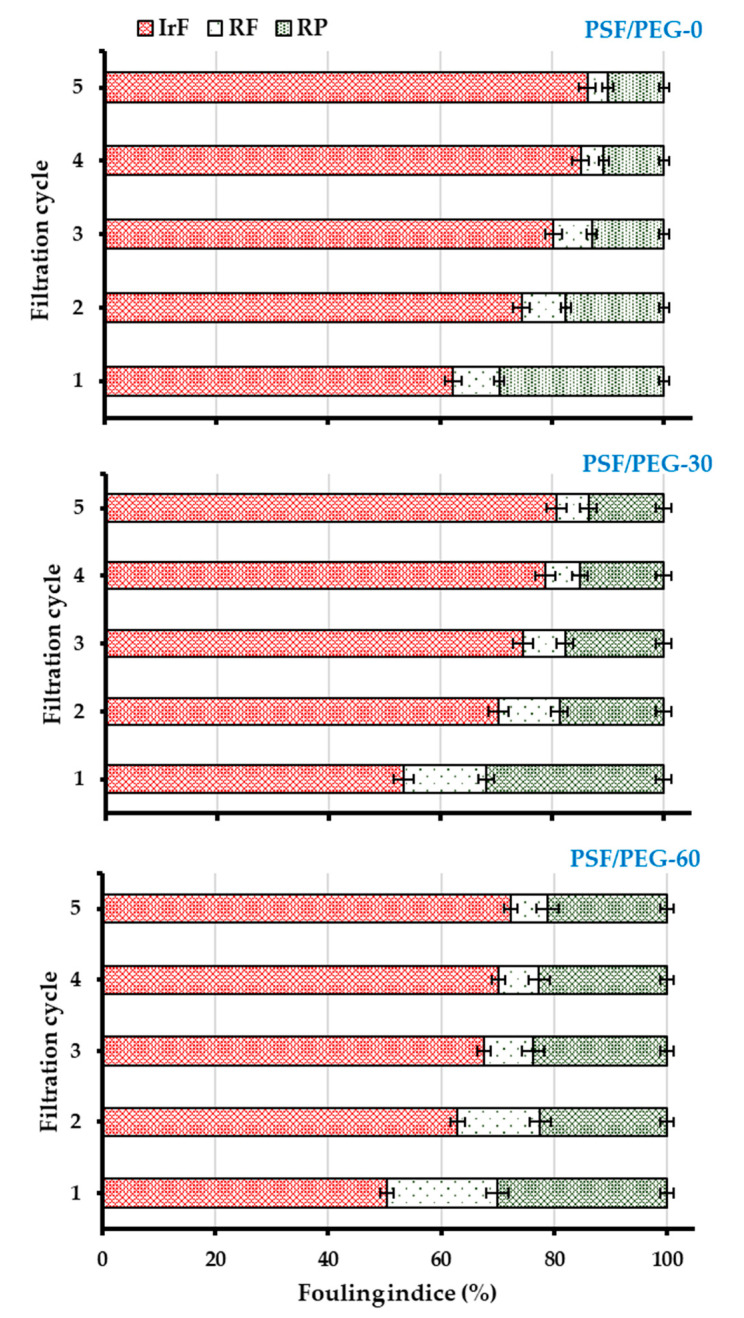
Membrane fouling analysis for oil/water emulsion filtration.

**Table 1 polymers-12-02519-t001:** Energy-dispersive X-ray spectroscopy surface elemental composition of the membranes.

Membrane	Composition (wt%)		
	C	O	S
PSF/PEG-0	70.58	24.64	4.78
PSF/PEG-30	69.40	25.86	4.74
PSF/PEG-60	69.02	26.26	4.72

**Table 2 polymers-12-02519-t002:** XPS surface elemental percentages of the membranes.

Membrane	Surface Elemental (mol%)			
	O 1s	C 1s	S 2p	O 1s/C 1s
PSF/PEG-0	16.99	81.49	1.52	0.208
PSF/PEG-60	17.96	80.16	1.88	0.229

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
