# Peer review of "Development of Polysulfone Membrane via Vapor-Induced Phase Separation for Oil/Water Emulsion Filtration"

_polymers, 2020, doi:10.3390/polym12112519_

Round 1

Reviewer 1 Report

The manuscript entitled: "Development of Polysulfone Membrane Via Vapor Induced Phase Separation for Oil/Water Emulsion Filtration" deals with new PSF membranes and their functionality for the successful separation of oil and water mixtures. A few issues need to be taken into account before acceptance for publication.

- For the water hydrophilicity characterization, it would be interesting to provide contact angle measurements. Is it possible?

- What is the effect of the ph on the membranes?

- You do not mention something about the fouling of the membrane or also threshold of the oil concentration in water that the membrane is still functional. Please comment on that.

- Please include a scalebar in SEM images and also provide the images larger

Author Response

Reviewer #1: The manuscript entitled: "Development of Polysulfone Membrane Via Vapor Induced Phase Separation for Oil/Water Emulsion Filtration" deals with new PSF membranes and their functionality for the successful separation of oil and water mixtures. A few issues need to be taken into account before acceptance for publication.

Comment 1: For the water hydrophilicity characterization, it would be interesting to provide contact angle measurements. Is it possible?

Response 1: The water contact angle of each membrane has been provided in Figure 5.

Location of changes: Figure 5, Line 285.

Comment 2: What is the effect of the ph on the membranes?

Response 2: Effect of oil/water emulsion pH on the membrane has not been studied in this research. The pH of oil/water emulsion used in this study is about normal (pH=6-8) and has no significant effect on the performances of polymeric membranes. To our best knowledge, the effect of pH is rarely reported in literatures treating normal oil/water emulsion using polymeric membranes. Unless special cases where the oil/water emulsion condition is harsh and therefore, must be considered during membrane selection and development.

Comment 3: You do not mention something about the fouling of the membrane or also

threshold of the oil concentration in water that the membrane is still functional. Please

comment on that.

Response 3: Membrane fouling analysis has been provided in sub-section 3.3 and Figure 11. Please refer to lines 370-391. While the oil rejection performances were provided in Figure 10, please refer to lines 366-370. Some discussion about the threshold concentration has also been included. It is anticipated that, due to severe fouling, the application of membrane for oil/water emulsion filtration is only applicable up to a certain threshold value. An economic tread-off is then expected between high concentration factor and the feasibility of the membrane. We plan of a follow-up study to address this issue.

Location of changes: Lines 146-147

Comment 4: - Please include a scalebar in SEM images and also provide the images larger

Response 4: The scalebar in in SEM images has been included. The figures have been

enlarged to show clearly the scale. Please refer to Figure 3.

Location of changes: Figure 3, Line 232

Reviewer 2 Report

The manuscript by Barambu and co-workers describe the vapour induced phase separation on the effect of the oil separation of polysulfone/polyethylene glycol membranes. The topic is of interest to both the membrane and polymer community and fits the scope of the journal. The manuscript has sufficient data and novelty to qualify publication in Polymers but there are several minor and major issues that need the authors serious attention before further consideration by the journal.

1) The experimental section lacks the characterization techniques for the oil-in-water emulsion. The emulsion size need to be determined and described.

2) The experimental section lacks the description of the methods used for the quantification of the oil content in the water used in equation 2 in the manuscript.

3) The surfactant to oil ratio of 1:9 (w/w) seems very high. What are the typical ratios in real scenarios? The authors need to explain in the manuscript whether such a high surfactant concentration is practically relevant. See section 2.2.

4) Figure 2 should include the operating parameters and equipment descriptors such as the flow rate of the pumps, flow rate of the permeate, membrane area, system volume, feed volume, collected permeate volume, applied pressure etc.

5) On what bases did the authors select Novatexx 24413 as the nonwoven support? A brief justification should be added to the manuscript, experimental section.

6) The authors state that the lithium chloride has a molecular weight of 42.38 kDa, which is incorrect (lines 123-124).

7) The recent achievement on oil-in-water produced water treatment using antifouling nanocomposite membranes should be acknowledged (DOI 10.1016/j.memsci.2020.118007).

8) The recent antifouling vapor-induced phase separation membranes also with polymer blending should be acknowledged (DOI 10.1016/j.memsci.2020.118256).

9) Usually a recirculation pump is used in the retentate loop during cross-flow filtration to minimize concentration polarization. It seems from the equipment schematic that this was not the case during the experiments carried out by the authors. Justification should be provided. What was the retenate/permeate flow rate ratio?

10) The authors write that 90 min filtrations were performed. In addition provide here the corresponding permeate volumes in parenthesis (line 169).

11) The FTIR spectra in figure 6 is not publishable. The resolution and pixilation are absolutely unacceptable. Also increase the size of the y axes for all spectra as they are very tiny. Plot the data appropriately. Currently the data cannot be interpreted.

12) Visible scale bars need to be added on the SEM images in figure 3. Enlarge the figure as in its current form it is too small to be able to see the fine features of the membranes.

13) Some critical assessment should be incorporated into the manuscript. What are the limitations and drawbacks of the proposed materials? What is the expected impact of the work?

14) Both the quotient (“x/y”) and negative exponent (“x y-1”) formats are used in the manuscript for units. Either of them should be used consistently, preferably the negative exponent format, which is recommended by the IUPAC.

15) Molecular weight is expressed as both g/mol and Da. Use only one of them for consistency.

Author Response

Reviewer #2: The manuscript by Barambu and co-workers describe the vapour induced phase separation on the effect of the oil separation of polysulfone/polyethylene glycol membranes. The topic is of interest to both the membrane and polymer community and fits the scope of the journal. The manuscript has sufficient data and novelty to qualify publication in Polymers but there are several minor and major issues that need the authors serious attention before further consideration by the journal.

Comment 1: The experimental section lacks the characterization techniques for the oil-inwater emulsion. The emulsion size need to be determined and described.

Response 1: The information on the oil/emulsion properties have been included in the revised manuscript. Discussions implied by the new data have also been included in the revised manuscript.

Location of changes: Lines 146-147, Lines 368-371.

Comment 2: The experimental section lacks the description of the methods used for the quantification of the oil content in the water used in equation 2 in the manuscript.

Response 2: Thank you for pointing out the missing details. Now, we have provided the required information in the revise manuscript.

Location of changes: Lines 185-187

Comment 3: The surfactant to oil ratio of 1:9 (w/w) seems very high. What are the typical ratios in real scenarios? The authors need to explain in the manuscript whether such a high surfactant concentration is practically relevant. See section 2.2.

Response 3: Thank you for the observation. It was typographical error, it was 1:99 and reference literature has been provided. Surfactant to oil ratio of 1:99 (w/w) was used because the research deals with oil/water emulsion not oil/water mixture. Therefore, appreciable amount of surfactant is required to keep the oil droplets in suspension and thus maintains its emulsion state. It is within the range recommended by the referenced literature provided.

Location of changes: Line 143

Comment 4: Figure 2 should include the operating parameters and equipment descriptors such as the flow rate of the pumps, flow rate of the permeate, membrane area, system volume, feed volume, collected permeate volume, applied pressure etc.

Response 4: The Figure 2 has been modified and the description of the equipment and

operating parameters have been provided.

Location of changes: Lines 163-166, Lines 168-170.

Comment 5: On what bases did the authors select Novatexx 24413 as the nonwoven

support? A brief justification should be added to the manuscript, experimental section.

Response 5: It was selected based on a reference that has been included in the revised

version. Also, there is a typo. It should be Novatexx 2413, instead of Novatexx 24413.

Location: Line 127

Comment 6: The authors state that the lithium chloride has a molecular weight of 42.38

kDa, which is incorrect (lines 123-124).

Response 6: Thank you for the observation. It has been corrected to 42.38 g/mol, the data provided by the manufacturer.

Location: Line 124

Comment 7: The recent achievement on oil-in-water produced water treatment using

antifouling nanocomposite membranes should be acknowledged (DOI

10.1016/j.memsci.2020.118007).

Response 7: It has been acknowledged.

Location: Line 58

Comment 8: The recent antifouling vapor-induced phase separation membranes also with polymer blending should be acknowledged (DOI 10.1016/j.memsci.2020.118256).

Response 8: It has been acknowledged.

Location: Line 59

Comment 9: Usually a recirculation pump is used in the retentate loop during cross-flow

filtration to minimize concentration polarization. It seems from the equipment schematic

that this was not the case during the experiments carried out by the authors. Justification

should be provided. What was the retenate/permeate flow rate ratio?

Response 9: We missed the detail on the method description. Actually, after measurement, the permeate was returned back to the feed. The details have been included in the revised manuscript.

Location: Line 173-174, Caption of Figure 2.

Comment 10: The authors write that 90 min filtrations were performed. In addition provide here the corresponding permeate volumes in parenthesis (line 169).

Response 10: This question is probably based on assumption that we did not performed fullrecirculation mode as mentioned in comment #9. The detailed filtration performances of the membranes have been provided in sub-section 3.2 and Figure 9. For each measurement (10 min of filtration cycle), the volume can be calculated back from the permeability data, membrane area and the transmembrane pressure.

Comment 11: The FTIR spectra in figure 6 is not publishable. The resolution and pixilation are absolutely unacceptable. Also increase the size of the y axes for all spectra as they are very tiny. Plot the data appropriately. Currently the data cannot be interpreted.

Response 11: The Figure has been modified.

Location: Figure 6, Line 304

Comment 12: Visible scale bars need to be added on the SEM images in figure 3. Enlarge the figure as in its current form it is too small to be able to see the fine features of the membranes.

Response 12: The Figure has been modified.

Location: Figure 3, Line 231-232

Comment 13: Some critical assessment should be incorporated into the manuscript. What are the limitations and drawbacks of the proposed materials? What is the expected impact of the work?

Response 13: Additional discussion on the limitation and the expected impact of the work have been included in the revised manuscript.

Location: Lines 389-393, Lines 403-409.

Comment 14: Both the quotient (“x/y”) and negative exponent (“x y-1”) formats are used in the manuscript for units. Either of them should be used consistently, preferably the negative exponent format, which is recommended by the IUPAC.

Response 14: The formats have been harmonized

Location: entire manuscript.

Comment 15: Molecular weight is expressed as both g/mol and Da. Use only one of them for consistency.

Response 15: The unit has been harmonized. Now, we use g/mol.

Location: Line 124

Round 2

Reviewer 1 Report

The manuscript can now be accepted for publication.

Reviewer 2 Report

The comments have been addressed and the manuscript is recommended for publication as is.